# Research on the Authenticity of Mutton Based on Machine Vision Technology

**DOI:** 10.3390/foods11223732

**Published:** 2022-11-21

**Authors:** Chunjuan Zhang, Dequan Zhang, Yuanyuan Su, Xiaochun Zheng, Shaobo Li, Li Chen

**Affiliations:** 1Institute of Food Science and Technology, Chinese Academy of Agricultural Sciences, Key Laboratory of Agro-Products Quality and Safety Control in Storage and Transport Process, Ministry of Agriculture and Rural Affairs, Beijing 100193, China; 2School of Food and Wine, Ningxia University, Yinchuan 750021, China

**Keywords:** machine vision technology, convolution neural network, livestock and poultry meat, adulterated minced mutton, authenticity identification

## Abstract

To realize the real-time automatic identification of adulterated minced mutton, a convolutional neural network (CNN) image recognition model of adulterated minced mutton was constructed. Images of mutton, duck, pork and chicken meat pieces, as well as prepared mutton adulterated with different proportions of duck, pork and chicken meat samples, were acquired by the laboratory’s self-built image acquisition system. Among all images were 960 images of different animal species and 1200 images of minced mutton adulterated with duck, pork and chicken. Additionally, 300 images of pure mutton and mutton adulterated with duck, pork and chicken were reacquired again for external validation. This study compared and analyzed the modeling effectiveness of six CNN models, AlexNet, GoogLeNet, ResNet-18, DarkNet-19, SqueezeNet and VGG-16, for different livestock and poultry meat pieces and adulterated mutton shape feature recognition. The results show that ResNet-18, GoogLeNet and DarkNet-19 models have the best learning effect and can identify different livestock and poultry meat pieces and adulterated minced mutton images more accurately, and the training accuracy of all three models reached more than 94%, among which the external validation accuracy of the optimal three models for adulterated minced mutton images reached more than 70%. Image learning based on a deep convolutional neural network (DCNN) model can identify different livestock meat pieces and adulterated mutton, providing technical support for the rapid and nondestructive identification of mutton authenticity.

## 1. Introduction

With the increase in income levels and changes in lifestyle, the meat consumption structure of Chinese residents has been transformed and upgraded, and the demand for mutton has continued to grow. At the same time, adulterations of mutton have occurred repeatedly, and there are frequent exposures to fake mutton [1,2,3], water-injected mutton [4], “lean” mutton [5,6] and other mutton adulteration events in the market, and the safety of mutton has become the focus of people’s attention. Due to the huge difference in the prices of different livestock and poultry meats, different forms of low-priced livestock and poultry meats adulterated with high-priced mutton have emerged in the pursuit of profit, such as unscrupulous elements adulterating low-cost duck, pork and chicken meat in mutton products, and some black-hearted merchants even directly use inedible mink meat, rat meat, fox meat and other counterfeits [7], making it hard for consumers to tell the difference, which not only causes damage to consumers’ interests and health [8] but also involves religious dietary taboos and other issues, so there is an urgent need for rapid, nondestructive testing technology in the market.

Machine vision technology uses optical systems and image processing equipment to simulate human vision, extracts information from the acquired target images and processes them to obtain the required information about the detected object and to analyze and judge it, which has the advantages of being nondestructive, fast, economical, consistent and objective [9,10]. Machine vision research focuses on image processing and classification and is relatively mature in the application of automatic meat grading and detection, mainly focusing on meat color [11], texture, marbling [12], acidity [13], tenderness [14] and freshness [15] detection, among others. Huang et al. [16] used images of pork from four different primal cuts as experimental data based on a computer vision method to achieve the automatic recognition and classification of pork primal cuts; the recognition result accuracy reached 94.47%, and the model achieved better results in the recognition of pork primal cuts and solved the problem of the difficult recognition of pork primal cuts. Ahmed et al. [17] studied RGB color imaging and a machine learning algorithm to detect plant and animal adulterants in minced meat with a proportion of 1–50%. The study was conducted to build a regression model to predict the number of adulterants, and the results showed that the linear discriminant classifier enhanced by the bagging ensemble performed the best, with an overall classification accuracy of 99.1% using all features to detect pure or adulterated samples.

Machine vision is a good basis for the authenticity detection of livestock and poultry meat pieces. Song et al. [18] applied smartphones to record videos with a series of different colors to detect adulterated minced beef by decomposing the videos into frames, extracting characteristic information and predicting the adulteration level with a partial least-squares regression model, which resulted in a prediction coefficient of 0.730–0.980. Zheng et al. [19] proposed thermal imaging combined with CNN for adulterated minced mutton detection, and the results showed that the accuracy of the qualitative CNN model on the validation and test sets was 99.97% and 99.99%, respectively, which obtained good results in the qualitative classification and quantitative prediction of mutton adulterated with pork. Previous studies based on machine vision to identify adulterated meat also use other auxiliary methods, the secondary processing of raw materials or manual feature extraction, while this study directly used machine vision technology to conduct experiments with different livestock and poultry meat blocks and adulterated mutton images as experimental data.

Therefore, this study took different livestock meat blocks and minced mutton meat adulterated with different proportions of duck, pork or chicken as the research object and used machine vision image recognition technology to identify the adulterated mutton images, aiming to provide a theoretical basis and new ideas for developing new technology for the authenticity identification of mutton.

## 2. Materials and Methods

### 2.1. Sample Preparation

#### 2.1.1. Sample Preparation of Different Livestock and Poultry Meat Pieces

The lamb leg, duck leg, pork leg and chicken breast were purchased from a supermarket in Haidian District, Beijing, placed in a foam box with ice packs and returned to the laboratory within 30 min. All meats were removed from the bones, skin, fat, blood, etc., and prepared by cutting them into irregular pieces with a mass of no more than 20 g.

#### 2.1.2. Sample Preparation of Minced Meat

Meats cut into small pieces were divided into 6 groups according to the proportion of duck (pork, chicken) meat added to the total sample (0%, 20%, 40%, 60%, 80% and 100% duck (pork, chicken) meat groups (as in Table 1)) and churned using a 350 W Meat Grinder for 15 s. Churning was repeated twice (with manual mixing at about 10 s intervals), and 60–70 g of the churned sample was taken and laid flat or compacted in a 90 mm diameter Petri dish.

#### 2.1.3. Sample Preparation for External Validation Sets

We re-purchased lamb leg meat, duck leg meat, pork leg meat and chicken breast meat at a supermarket following the experimental process of 2.1.2 to produce 6 samples of each doping ratio in order to ensure the consistency of external modeling data, and 30 samples of pure lamb meat were produced separately. A total of 120 samples were produced for external validation experiments; front image and reverse images of each sample were collected. The specific sampling quantities are shown in Table 2.

### 2.2. Machine Vision Image Acquisition and Calibration

The image acquisition device mainly consists of a hardware device and a software device, where the hardware device includes a CMOS camera, camera obscura, light source and computer, as shown in Figure 1. The camera and lens parameters are shown in Table 3, and the camera was vertically positioned 12 cm above the stage. The camera was connected to the computer via Daheng Galaxy Viewer (x64) software, and the images were acquired remotely.

The computer hardware configuration: Windows 10 Home Edition 64-bit operating system, CPU with ADATA DDR4 3200 MHz, main frequency 2.60 GHz; Graphic Processing Unit (GPU) with Nvidia GeForce RTX 3060 and 12 GB video memory. The experiments were completed in the MatlabR2021a Deep Learning Tool software environment.

Firstly, the camera was white-calibrated, and the white balance coefficient of the camera was fixed. After calibration, the prepared minced meat samples were placed on the carrier table, and the height of the carrier table was adjusted to ensure that the minced meat/culture dish appeared in the camera’s field of view. Then, the sensitivity was determined by adjusting the size of the camera exposure rate, the images were acquired by fixing the tuned parameters, and two images were acquired for each sample.

### 2.3. Research on Classification Network Based on Convolutional Neural Network

#### 2.3.1. Convolutional Neural Network

CNN is a kind of Feedforward Neural Network (FNN) with convolution calculations and a deep structure. It can extract effective features of input information autonomously due to the inclusion of multiple convolution layers and perform progressive abstraction layer by layer. After a multi-layer transformation, a deep network can transform the original image into a high level of abstraction. A typical CNN is shown in Figure 2, which is divided into a convolutional layer, activation function, pooling layer and fully connected layer. The convolutional layer is the core of the CNN, which is responsible for most of the computational work, thus extracting features, reducing the number of parameters to be trained and reducing the complexity of the deep network; the activation function is a nonlinear factor added to improve the expression ability of image features when the CNN is running, and the linear model has insufficient ability to represent the image features. The activation function is a nonlinear factor added to improve the representation of image features when the linear model is not sufficient for the representation of image features when the CNN is run. The RELU activation function can effectively solve the gradient dissipation problem, which can shorten the training time, and it is also multi-level to show the representation effect of the model [20]. The pooling operation is essentially a process of extracting statistical information, and the pooling layer, also known as the downsampling layer, usually exists after the convolution layer, which performs the feature selection and dimensionality reduction of the input features through a series of local nonlinear operations to reduce the model parameters and improve the network’s ability to resist distortions, such as the translation and rotation of the input image; the fully connected layer is used to combine the extracted features, and the overall characteristics of the image can be seen by locally composing the global, appearing after multiple convolutional and pooling layers are stacked alternately, and the extracted features are further downscaled to input the features into the softmax layer.

#### 2.3.2. AlexNet Network Structure

AlexNet is a sensational convolutional neural network that was first trained by Alex Krizhevsky on a GPU in 2012, who won the ImageNet 2012 image recognition challenge by a huge margin, and convolutional neural networks in the field of image recognition have been growing rapidly since then. The model groups mainly contain one input layer, one output layer, three convolutional layers, three pooling layers and two fully connected layers. The difference between this method and ordinary CNNs is that the activation function of the network is changed from Sigmoid to ReLU, which helps the neural network better solve complex nonlinear problems; the local response normalization algorithm (LRN) is added after the pooling layer, and a hidden dropout layer is added before the fully connected layer to improve the generalization ability of the whole network.

#### 2.3.3. VGG-16 Network Structure

VGG is a DCNN developed by the Oxford University computer vision team together with researchers at Google DeepMind, inheriting part of the structure of AlexNet, but with about three times the number of parameters of AlexNet. VGG-16 mainly contains one input layer, one output layer, thirteen convolutional layers, five pooling layers and three fully connected layers, and as the number of layers deepens, the network width becomes smaller, while the number of channels increases.

#### 2.3.4. SqueezeNet Network Structure

In 2016, Forrest N. Iandola et al. proposed SqueezeNet, a lightweight network model, which replaces some of the 3 × 3 convolutional kernels with 1 × 1 convolutional kernels, reducing the training parameters of the CNN model and compressing the size of the network model compared to AlexNet, which uses only 3 × 3 convolution. At the same time, the accuracy of recognition is greatly improved by delaying the pooling layer to obtain a larger feature image. The SqueezeNet network structure mainly consists of a convolutional layer, pooling layer, Fire module and ReLU activation layer. Among them, the Fire module is the core part of the SqueezeNet network.

#### 2.3.5. GoogLeNet Network Structure

GoogLeNet is a new network model proposed by the Google team in 2014, which optimizes the structure of the network, resulting in a significant reduction in the number of parameters and computation, with 22 layers. GoogLeNet has deeper layers than VGG net, but the number of parameters is greatly reduced, and the classification accuracy on the ImageNet dataset is much higher than the previous network model.

#### 2.3.6. ResNet-18 Network Structure

The Residual Network (ResNet) model was proposed by He et al. [21] in 2015, and the main difference from other networks is that the idea of the residual is introduced in the CNN to increase the depth of the network, mainly by adding shortcut connections to update the gradient connected by jumping layers, which solves the problem that after the network becomes deeper, the network weights of the previous layers cannot be updated, which leads to the disappearance of the gradient, and thus improves the network image feature extraction ability, which is widely used for all kinds of image recognition. Therefore, the number indicates the depth of the network, and the 18 in ResNet-18 refers to the 18 layers with weights, including the convolutional layer and the fully connected layer, excluding the pooling layer and the BN layer.

#### 2.3.7. DarkNet-19 Network Structure

DarkNet is an open-source neural network framework based on C language and CUDA that supports the Linux operating system, supports CPU and GPU operations, supports OpenCV processing image operations, etc. It has the characteristics of simple installation, a small size and fast speed. The design of this CNN network classification model is based on the advantages of many previous successful CNN network classification models and has been used for the classification of various types of images, indicating that Darknet-19 is a reasonable and well-performing CNN network classification model.

### 2.4. Model Construction and Testing Process

Based on the principle of shared parameter migration learning, this study improved six models, which were AlexNet, GoogLeNet, ResNet-18, DarkNet-19, SqueezeNet and VGG-16, and used the sample dataset for training and learning, mainly replacing the image input layer and the final fully connected layer, softmax layer and classification layer of the model. The output size of the fully connected layer is modified to the corresponding number of classifications, and other parameters are kept unchanged. Finally, by fine-tuning the training parameters, the recognition of different animal and poultry meat and adulterated minced meat images is achieved.

The flow chart of training and testing of the dataset is shown in Figure 3.

The training set was trained according to the process in Figure 3a; after each training epoch, the validation set was tested according to Figure 3b, the accuracy on the test set was recorded, and the model was saved.

### 2.5. Judgment Indicators

This study used accuracy (%), loss (%), model size (MB) and training time (min) to evaluate the performance of all classification models. The accuracy rate is the training accuracy, which refers to the correct image ratio in all of the recognized images, which can reflect the training effect of the model. The calculation is as follows:(1)$$\mathrm{Accuracy}=\frac{\mathrm{TP}+\mathrm{TN}}{\mathrm{TP}+\mathrm{TN}+\mathrm{FP}+\mathrm{FN}}\times 100\%.$$
where TP, FP, FN and TN are positive samples, negative samples, positive samples and negative samples predicted by the model, respectively.

The loss value can estimate the degree of deviation between the predicted and true values of the model during the training and testing process and can determine whether the training process of the model converges, which is calculated as:(2)$$\mathrm{Loss}=\left\{\frac{e^{f}\mathrm{yi}}{\sum_{n=1}^{K}e^{f}n}\right\}$$
where yi is the label corresponding to the ith sample, f is the model output function, n is the summation variable, and K is the total number of samples.

## 3. Results

### 3.1. Grouping of Datasets

Data augmentation is the process of generating more samples from existing data by introducing operations, which serves to expand the effective training samples and prevent the model from learning inadequately due to too little training data, where the results obtained from training cannot meet expectations. In this study, the data were enhanced by flipping, cropping, shifting, etc. The numbers of original images and data augmentation images collected in this study are shown in Table 4: 960 images of different livestock and poultry meat blocks and 1200 images of minced mutton adulterated with duck, pork and chicken were obtained, and 300 images of the external validation set of adulterated minced mutton were re-collected.

### 3.2. Model Learning Parameter Determination

#### 3.2.1. Learning Rate Determination

The learning rate determines the gradient descent rate during the training process. Too large a learning rate will increase the amplitude of the gradient iterations and cause the model to miss the optimal solution; too small a learning rate will reduce the convergence rate of the iterations and make it difficult for the model to find the optimal solution. Three sets of learning rates (0.01, 0.001 and 0.0001) were selected to determine the optimal parameters with a fixed small batch value of 64, and the results of two lightweight models of SqueezeNet and GoogLeNet are shown in Figure 4. When the learning rate is 0.001, the training, validation and testing accuracy of both models are the highest, where the accuracy of the SqueezeNet model was 98.12%, 91.88% and 93.13% for the training set, validation set and testing set, respectively; the accuracy of the GoogLeNet model is 100%, 86.25% and 85.63% for the training, validation and test sets, respectively.

#### 3.2.2. Mini-Batch Value Determination

A mini-batch refers to dividing the training set into several smaller training sets for multi-stage training, and a suitable small batch can accelerate the training speed and avoid the local optimum problem. With a fixed learning rate value of 0.001 and three sets of small batch values (32, 64 and 128) chosen according to the sample size of the test data to determine the optimal parameters, the results of SqueezeNet and GoogLeNet models are shown in Figure 5. The best results were obtained for the model with a small batch size of 32, in which the accuracy rates of the SqueezeNet model were 96.88%, 99.38% and 98.75% for the training set, validation set and test set, respectively, and the accuracy rates of the GoogLeNet model were 100%, 98.13% and 97.50% for the training set, validation set and test set, respectively. The above findings are consistent with those of Junjie Wan [22] and Minchong Zheng, et al. [19]. Both indicate that the highest model training accuracy is achieved when the learning rate is 0.001. Therefore, 0.001 and 32 were chosen as the learning rate and small batch value for model training in this study.

### 3.3. Research on the Image Recognition Method of Different Livestock and Poultry Meat Pieces Based on Deep Convolutional Neural Network

Training accuracy refers to the training accuracy when the network is trained and gradually plateaus as the number of iterations increases. The higher the training accuracy, the better the network performance. The training accuracy curves of the six transfer learning models are shown in Figure 6a, where the training accuracy curves of the six models are approximately the same, and the training accuracy is more than 90%. However, overall, ResNet-18 converges the fastest, and the slowest models are AlexNet and VGG-16, although AlexNet fluctuates a bit more. When the number of model iterations is 100, the training accuracy of ResNet-18 is the first to reach 100%; the training accuracy stabilizes in the next 500 iterations, and finally, the training accuracy is 100%. The training loss curve in Figure 6b shows that the smallest training loss value of ResNet-18 was 0.0009. The training results indicate that the training accuracy and training loss of the ResNet-18 network model are optimal compared with the other five models in this study, which indicates that this model has a better learning ability for different livestock and poultry meat blocks.

The training time of the six models and the final validation accuracy of the models are shown in Table 5, where the training time of the models reflects the time complexity of the network structure, and the training time of the network is also a comprehensive index to evaluate the network performance. The validation accuracy can be used to evaluate the generalization ability of the model, and the higher the validation accuracy, the better the performance of the network. Among the six training network models, except for VGG-16, the training time of the remaining five models is basically the same, among which the AlexNet model has the shortest training time and the highest validation accuracy, followed by SqueezeNet and ResNet-18 models, whose training accuracy is 99.375%. In general, the more layers of the network and the more parameters of the network model, the larger the amount of data needed to train the model and the longer the training time required. The time is also the longest.

The classification confusion matrix of different model test sets for different livestock and poultry meat block samples is shown in Figure 7. The results showed that the ResNet-18 and SqueezeNet network models had the highest test accuracy. Only one pork sample was wrongly predicted as chicken in the ResNet-18 network model, and the test accuracy was 99.375%; one mutton sample and one pork sample were wrongly predicted as duck in the SqueezeNet network model, and the test accuracy was 98.75%. The AlexNet model had the lowest test accuracy, in which six mutton blocks were incorrectly predicted as pork, three pork blocks were incorrectly predicted as duck, and two were incorrectly predicted as chicken, with an accuracy of 94.375%.

By comparing the migration learning results of the six models with the same training parameters, it was found that the ResNet-18 model outperformed the other five models in terms of training accuracy (100%), validation accuracy (99.375%) and training loss (0.0009) values, with good recognition results. The results suggest that ResNet-18 is more suitable for this study model.

### 3.4. Research on Mutton and Duck-, Pork- and Chicken-Adulterated Minced Mutton Image Recognition Method Based on Deep Convolutional Neural Network

Image transfer learning was performed on mutton and mutton samples adulterated with different meat sources as a whole. The training accuracy curves and loss curves of the six models are shown in Figure 8. The training accuracy curves of the six models are roughly the same, and the training accuracy reaches more than 93%, but it is obvious from the figure that the ResNet-18, GoogLeNet and DarkNet-19 models have the fastest convergence speed, among which ResNet-18 and DarkNet-19 models have the highest training accuracy of 100% and 100%, respectively, and their corresponding loss values are the smallest, 0.004 and 0.02, respectively; the slowest convergence speed was obtained with VGG-16, and the lowest training accuracy and the largest loss value were obtained with the SqueezeNet models, with 93.29% and 0.149, respectively.

The training time, model validation accuracy and final model size of the six models are shown in Figure 9. Among the six training network models, the VGG-16 model took the longest time, and the remaining five models took basically the same time; the validation accuracy of DarkNet-19, ResNet-18 and GoogLeNet models were the highest at 98.75% and 98.33%, and the validation accuracy of the SqueezeNet model was the lowest at 67.08%.

The classification confusion matrix of different model test sets for mutton and mutton species adulterated with different meat source samples is shown in Figure 10. The results of DarkNet-19, ResNet-18 and GoogLeNet, three models that tested with high accuracy, were 98.33%, 98.33% and 96.67%, respectively, where DarkNet-19 and ResNet-18 models both classified 5 samples incorrectly, and the DarkNet-19 model predicted 10 samples incorrectly. The highest test error rate was achieved by the AlexNet model, which predicted a total of 20 samples incorrectly, with a correct test rate of only 93.33%.

### 3.5. External Validation Results of Mutton Authenticity Discrimination Model Based on Machine Vision Technology

Three hundred images were re-captured for the external validation of GoogLeNet, ResNet-18 and DarkNet-19, which were the better of the six CNN models trained for pure mutton and minced mutton adulterated with pork, duck and chicken samples, and the results are shown in Table 6. The external validation of the DarkNet-19 model was better, with a validation accuracy of 75.33%, while the GoogLeNet and ResNet-18 models had a validation accuracy of 73.67% and 71.67%, respectively.

## 4. Discussion

With the increased emphasis on food quality and the development of optical, computer and artificial intelligence technologies, color-based food inspection techniques are also developing rapidly, and because of their advantages, such as nondestructive, rapid and sustainable tracking [23], machine vision imaging techniques have received great attention in the detection of the quality attributes of agricultural products, including meat and meat products. Machine vision techniques have the potential to replace human vision and image perception in meat quality assessment and safety assurance. Image recognition is an important branch of machine vision technology: i.e., it is used to distinguish between different classes of images. CNN is one of the best algorithms to accomplish the image recognition task, and by training and testing the input samples and extracting features from simple to deep to distinguish the samples, the image classification error can be reduced, and a high recognition rate can be obtained [24].

External features such as color and texture are also important indicators for evaluating the freshness of meat, which largely determines the consumer’s desire to buy. Fresh meat has a bright red color, clear texture, dry meat and no mucus, while spoiled meat has a dark red color or even turns brown or green, a blurred texture and abundant mucus [25]. Meat color is produced and determined by myoglobin and hemoglobin in terms of chemical composition. The content of myoglobin is influenced by the type of animal, muscle part, degree of exercise, age and sex, and the meat color varies among various types of animals [26]. The texture of meat is usually more intuitive for observing the structural state of meat, and the texture structure of different livestock and poultry meat varies. Computer vision technology is based on the principle of human vision, and images are used to obtain information on external characteristics, such as meat color and texture, and to discriminate and grade various types of meat. Therefore, these physical characteristics of meat may help in the detection of meat adulteration. In this study, based on the color difference and texture structure of different livestock and poultry meat blocks for different livestock and poultry meat images’ identification, the results of six models tested with an accuracy rate of 94% or more; for the recognition training of color recognition based on adulterated minced mutton images, the final external verification accuracy of its optimal model reached 75%. The reason for the low accuracy of external verification is that machine vision is limited to identifying and extracting external image features or quality factors (e.g., color, size and surface structure) of samples [27,28,29,30] and cannot yet take into account the chemical composition and internal quality characteristics of food products. Food color is an important basis for determining food quality, but the current focus of detection is only on the color itself, and often, the measurement results are rather limited [31]. This further explains that the reason for the poor accuracy of external validation in this study is because of the change in sample color. The internal factors that affect the color change of meat samples are pH, ambient temperature, high or low oxygen content in the air, microorganisms and metal ions in the meat; in addition, the color change of meat samples is also affected by light, the type of meat and the outer packaging of meat, so it leads to low accuracy of external validation. To utilize color information in more depth, new algorithms can be developed to associate color with other quality parameters, and after training to improve the accuracy of the judgment, color parameters as well as other obtained parameters can be used to characterize other quality parameters [23].

The above study shows that from the point of view of machine vision, the difference between adulterated and unadulterated samples mainly lies in the color difference or texture structure, and the combination of machine vision technology and chemometric methods is an effective method to identify adulterated mutton, not only for different livestock and poultry meats but also for adulterated meat, but still based on many complex images’ processing and a series of processes, such as model building, to identify it. So, the development of rapid, portable detection equipment has become particularly important, and in the current market situation in China, the authenticity of meat has always been the focus of consumer complaints and social concerns. Meat authenticity detection technology is a re-innovation, from the traditional physical and chemical identification to the application of new nondestructive technology today, no doubt to add convenience to the authenticity of meat detection. However, existing nondestructive testing technology still has certain limitations; for example, it cannot be applied to real-time on-site monitoring, so it is necessary to establish real-time meat authenticity identification methods based on consumer applications, such as the development of smartphone apps, applying its advantages of high portability, simple use and free use of places to achieve the consumer identification of mutton authenticity, which has good application value and wide market prospects. Of course, in the future, there is still a need for targeted mapping data fusion and attempts to develop artificial intelligence deep learning algorithms, deep excavation indicating the fingerprint characteristics of various types of meat and screening out data information applicable to the detection of multiple or certain types of adulteration. Finally, it is still necessary to optimize the application conditions, establish application methods, conduct research and development of supporting large-scale equipment or small portable equipment, and ultimately provide key technologies and equipment for the rapid real-time detection and identification of the phenomenon of meat adulteration in China to enhance the level of supervision.

## 5. Conclusions

In this study, a migration-learning-based CNN for adulterated mutton image recognition is proposed. A network model suitable for adulterated mutton image recognition was constructed by the migration comparison learning of six models, namely, AlexNet, SqueezeNet, GoogLeNet, ResNet-18, DarkNet-19 and VGG-16. The validation accuracy of the ResNet-18 model built based on DCNN reaches 99%, and the external validation accuracy of adulterated mutton reaches 75%, which indicates that the ResNet-18 model has a better learning ability for mutton image authenticity recognition and is more suitable for this study. The results of this study provide a reference for further research in the future and also provide some technical support for the subsequent development of a visual adulterated mutton classification and recognition system.

## Figures and Tables

**Figure 1 foods-11-03732-f001:**
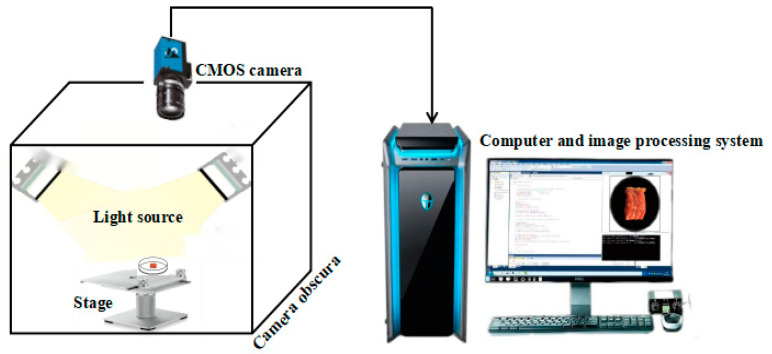
Image acquisition device.

**Figure 2 foods-11-03732-f002:**
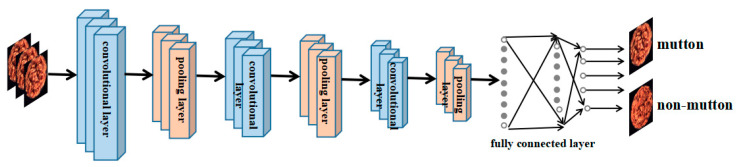
CNN model for mutton authenticity recognition.

**Figure 3 foods-11-03732-f003:**
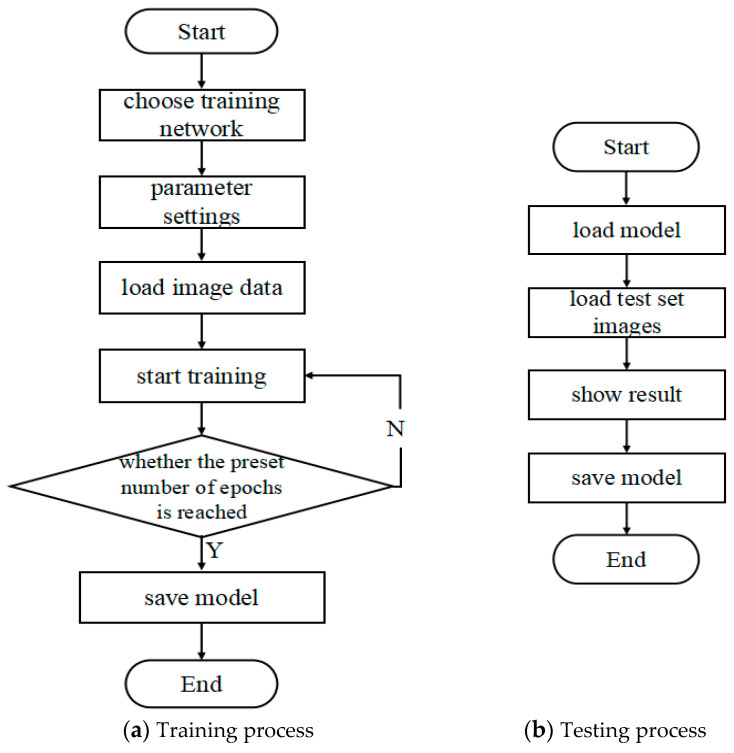
Model testing process.

**Figure 4 foods-11-03732-f004:**
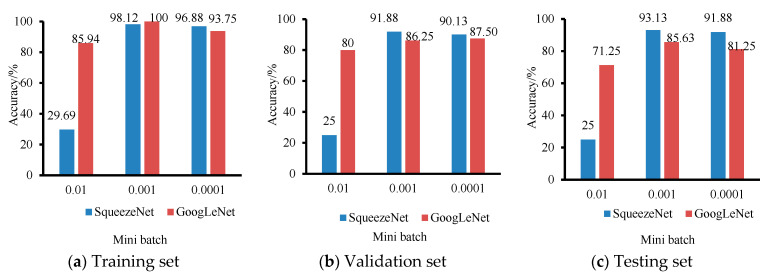
The results of SqueezeNet and GoogLeNet models with a mini-batch of 64 and three groups of learning rates (0.01, 0.001 and 0.0001).

**Figure 5 foods-11-03732-f005:**
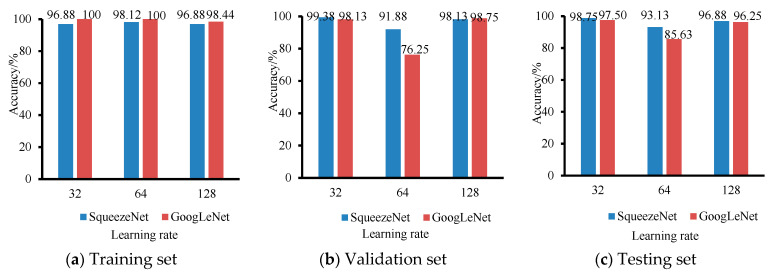
The results of SqueezeNet and GoogLeNet models with the learning rates of 0.001 and three groups of mini-batches (32, 64 and 128).

**Figure 6 foods-11-03732-f006:**
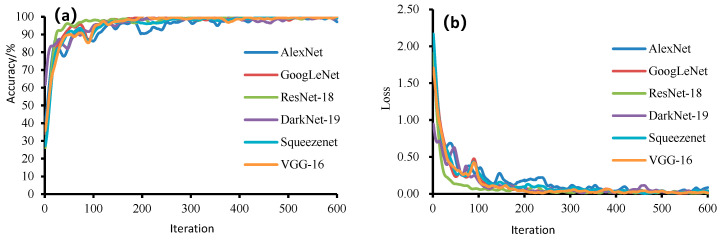
Training accuracy curve (**a**) and loss curve (**b**) of six models.

**Figure 7 foods-11-03732-f007:**
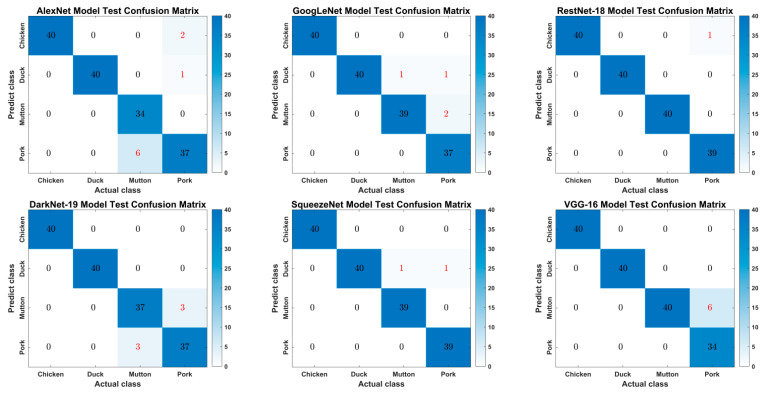
Matrix of 6 types of models’ test set classification confusion.

**Figure 8 foods-11-03732-f008:**
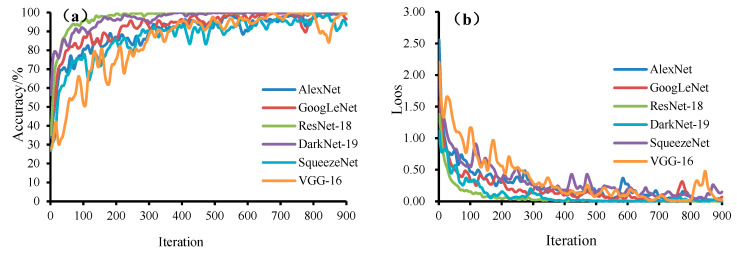
Training accuracy curve (**a**) and loss curve (**b**) of six models.

**Figure 9 foods-11-03732-f009:**
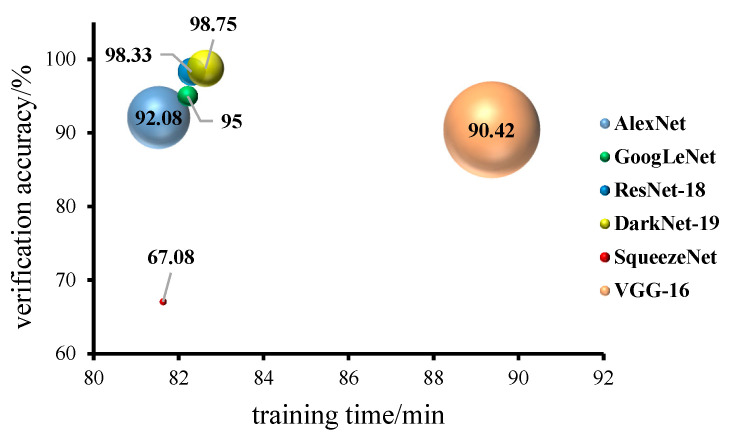
Comparison of training duration and validation accuracy of six models. (The size of the bubble indicates the size of the model after training is completed, MB; the model training time is the time used for model training, min; the value on the bubble indicates the validation accuracy, %; the same meanings apply below).

**Figure 10 foods-11-03732-f010:**
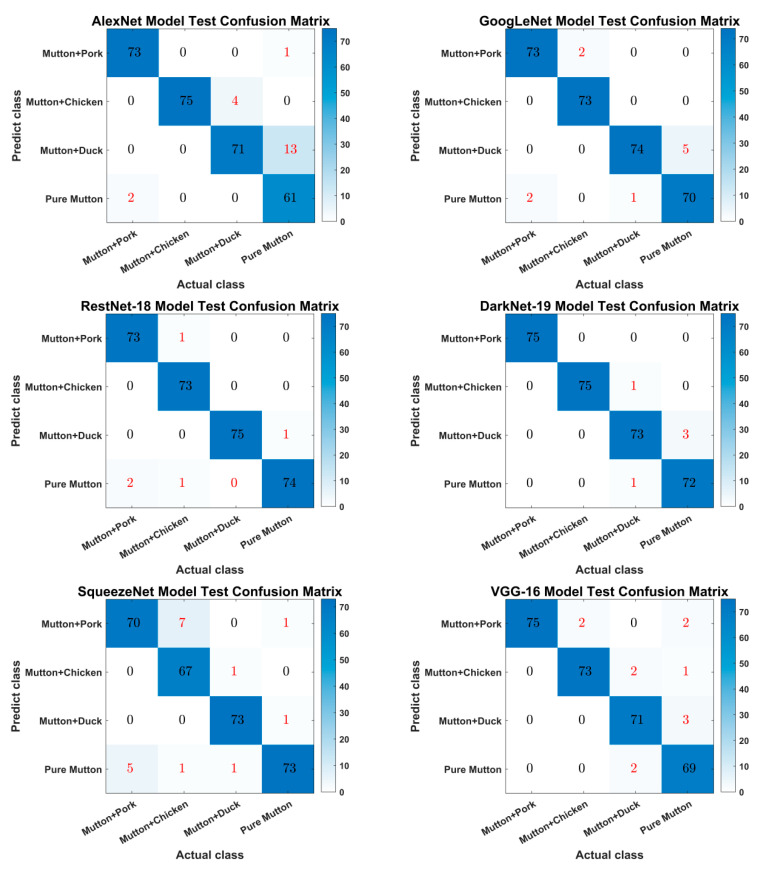
Matrix of 6 types of model test set classification confusion.

**Table 1 foods-11-03732-t001:** Composition of different livestock and poultry meat samples and adulterated meat samples.

Grouping Type	Group Name	Composition of the Samples
Adulterated minced meat	Pure mutton	0% duck, 0% duck, 0% duck
	Minced mutton adulterated with duck	0% duck, 20% duck, 40% duck, 60% duck, 80% duck, 100% duck
	Minced mutton adulterated with pork	0% pork, 20% pork, 40% pork, 60% pork, 80% pork, 100% pork
	Minced mutton adulterated with chicken	0% chicken, 20% chicken, 40% chicken, 60% chicken, 80% chicken, 100% chicken

**Table 2 foods-11-03732-t002:** External Validation Experiment Dataset.

Group	Number of Samples					Total	Acquisition of Images: Number of Images
	20%	40%	60%	80%	100%		
Minced mutton adulterated with duck	6	6	6	6	6	30	60
Minced mutton adulterated with pork	6	6	6	6	6	30	60
Minced mutton adulterated with chicken	6	6	6	6	6	30	60
Pure mutton	-					30	60
**Total**						120	240

**Table 3 foods-11-03732-t003:** MER-2000-5GC-P camera and HN-1226-20M-C1/1X lens parameters.

Images	Parameters	Parameter Values	Advantages
** 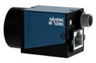 **	Model	MER-2000-5GC-P	High definition, low noise, compact design, easy to install and use, and suitable for industrial testing, medical treatment, scientific research, education, security and other fields
Resolution	5496 (H) × 3672 (V)
Frame rate	5 fps
Sensor type	1” Sony IMX183 exposure CMOS
	Cell size	2.4 μm × 2.4 μm
Image data format	Bayer RG8/Bayer RG12
Signal-to-noise ratio	45 db
Data interface	Fast ethernet or gigabit ethernet (100 Mbit/s)
Operation temperature	0–45 °C
Operation humidity	10–80%
Weight	75 g
** 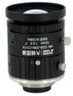 **	Model Chart size	HN-1226-20M-C1/1X 1”	Specifically designed for machine vision applications, compact, convenient, low distortion, high shock resistance, uniform illumination and sufficient brightness at the corners of the screen
Focal length (mm) Maximum aperture ratio Maximum imaging size Aperture range Working distance	12 1:2.6 12.8 × 9.6 (Φ16) F2.6–F16.0 0.1 m–Inf.
Back focal length	10.6 mm
Weight Operation temperature	98.4 g −10–50 °C

**Table 4 foods-11-03732-t004:** The samples of original number and number after augmentation of different breeds and adulterated meat.

Classification Type	Classification		Raw Data Volume	After Data Augmentation	Training Set	Internal Test Set	Total
Different livestock and poultry meat	Mutton		120	240	200	40	960
	Duck		120	240	200	40	
	Pork		120	240	200	40	
	Chicken		120	240	200	40	
Adulterated minced meat	Pure mutton	0% duck, 0% pork, 0% chicken	150	300	225	75	1200
	Minced mutton adulterated with duck	20–100% duck	150	300	225	75	
	Minced mutton adulterated with pork	20–100% pork	150	300	225	75	
	Minced mutton adulterated with chicken	20–100% chicken	150	300	225	75	
External validation dataset	Pure mutton	Pure mutton	60	120	external test set	75	300
	minced mutton adulterated with duck	20–100% duck	60	120		75	
	Minced mutton adulterated with pork	20–100% pork	60	120		75	
	Minced mutton adulterated with chicken	20–100% chicken	60	120		75	

**Table 5 foods-11-03732-t005:** Comparison of training duration and validation accuracy of six training network models.

Network Model	Training Time	Verification Accuracy%
AlexNet	42 min 7 s	100%
GoogLeNet	44 min 33 s	98.125%
ResNet-18	42 min 28 s	99.375%
DarkNet-19	43 min 22 s	98.75%
SqueezeNet	42 min 16 s	99.375%
VGG-16	49 min 48 s	96.875%

**Table 6 foods-11-03732-t006:** External validation results of the optimal CNN model for samples of mutton and minced mutton adulterated with duck, pork and chicken.

Model	Number	Real Category	Number of Correct External Validations				Correct Rate/%
			Minced Mutton Adulterated with Pork	Minced Mutton Adulterated with Chicken	Minced Mutton Adulterated with Duck	Pure Mutton	
GoogLeNet	300	Minced mutton adulterated with pork	40	2	0	0	73.67
		Minced mutton adulterated with chicken	0	72	0	0	
		Minced mutton adulterated with duck	0	0	28	04	
		Pure mutton	35	1	47	75	
ResNet-18	300	Minced mutton adulterated with pork	40	2	0	0	71.67
		Minced mutton adulterated with chicken	0	72	0	0	
		Minced mutton adulterated with duck	0	0	28	0	
		Pure mutton	35	1	47	75	
DarkNet-19	300	Minced mutton adulterated with pork	40	0	0	0	75.33
		Minced mutton adulterated with chicken	0	75	0	0	
		Minced mutton adulterated with duck	0	0	36	0	
		Pure mutton	35	0	39	75	

## Data Availability

Data is contained within the article.

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
