# Peer review of "Research on the Authenticity of Mutton Based on Machine Vision Technology"

_foods, 2022, doi:10.3390/foods11223732_

Round 1

Reviewer 1 Report

The authors propose a Deep Learning (DL) system based on a Convolutional Neural Network (CNN) to identify adulterated mutton meat, that is a growing issue in China. Different state-of-the-art CNN architectures are tested on the problem, with good results delivered.

Overall, the methodology seems to make sense. The experimental analysis looks reasonable, and the results are good enough. I am surprised by the extreme drop in accuracy when tested on new images. It's interesting to know, though, and the explanation seems to make sense.

Some points for future works: 

- have you considered using explainable AI techniques to look into the filters, and recognize which parts of the image the CNN is focusing on? It could deliver more insights on the classification process.

- I have read other works where they obtained good results with Near Infra-Red (NIR) spectrometry to detect adulterated food products. It's the same people that worked on this: https://ieeexplore.ieee.org/document/7501348 (but the work was with fish before processing); and later they moved to NIR, obtaining good results on processed fish filets. I am not sure they ever published the NIR-related work, but it was in a chapter of the Ph.D. Thesis of Francesco Rossi, "Computer-Aided Technologies for Food Risk Assessment". I think it may be accessible through here: https://123dok.org/document/q7wxgx5v-politecnico-di-torino-repository-istituzionale.html or here: https://iris.polito.it/handle/11583/2714103#.XnHntC2ZPOQ

- while DL and CNN approaches are powerful, they often have trouble generalizing to never-seen-before samples. In your case, it would be interesting to know what happens if you just train the network using only two types of adultered meat (for example, duck and pork), and use the third one (e.g. chicken) as a test. This experiment could give some insight in what would happen if the network was tested on a new type of adulteration (in the introduction you mention mink or fox, for example): would it be able to recognize it? Or can it only recognize types of meat observed during training? Should the generalization be poor, you could try extending your training set. 

There is a typo in Table 4, "Affter Data augmentation" (extra 'f')

In Table 5, by "Verify accuracy%" you mean "validation accuracy%", right?

Author Response

Thank you very much for your reasonable comments and questions on "foods-1987681 Research on the authenticity of mutton based on machine vision technology". We have carefully revised the text as requested, as shown in "foods-1987681- Revised version", with all changes highlighted in red. The point to point response is available in the word document "point to point response" below.

Reviewer 2 Report

The authors undertook to analyze the authenticity of mutton based on machine learning and machine vision from photo samples representing actual and adulterated mutton. In particular, they proposed migration learning-based CNN for adulterated mutton.

The introduction is short but legible and completely covers the issues faced by the authors of the work. The authors cited several existing works on similar problems but different types of meat and used various features and recognition methods for the methods presented in the literature. The authors noted methods based on imaging as well as thermal imaging.

I cannot judge adequately prepared samples as I am not competent in preparing food samples. However, I assume that authentic and adulterated minced meat samples were prepared objectively, reflecting the samples in actual sales. Using purchased samples with counterfeit meat would be ideal, but it would not be easy to complete such an experiment quickly. Because the experiments were repeated many times with meat from other purchases, it can be concluded that they were pretty objective. Only then can we consider the experiment accurate.

Considering the above assumption, the authors have prepared a test stand to experiment with the same lighting conditions. This approach allows you to perform image acquisition objectively, maintaining the same conditions. How long does it take to take 240 photos of the prepared samples, and does the prepared meat not change its color or structure over time?

Suppose the structure of the meat does not change during the acquisition of the samples. In that case, it can be assumed that the identical acquisition conditions have been met and that the meat samples have been objectively recorded in a series of photos.

The authors concluded that the internal factors that affect the color change of meat samples are pH, ambient temperature, high or low oxygen content in the air, microorganisms, and metal ions in the meat, as well as color change.

Another question is, does the thickness or variability of the outer layer of the same sample affect the variability of the images? How did the authors deal with this problem, were the samples smooth, or did the test bench have multi-directional illumination that could eliminate the potential shadows appearing in the case of uneven meat surfaces?

Building the model and conducting training and testing is proper, and the selected networks' structures are diverse. The additional data generation was also performed correctly. The methodology for choosing the optimal model was also adopted correctly.

By comparing the migration learning results of the six models with the same training parameters, the authors have found that the ResNet-18 model outperformed the other five models in training, validation accuracy, and training loss values, with good recognition results. The authors have chosen ResNet-18 as more suitable for the presented study model.

The authors also performed external validation and rightly concluded that the reason for the low accuracy of external verification is that machine vision is limited to identifying and extracting external image characteristics or quality factors and does not consider the chemical composition and internal quality characteristics of food products.

The authors also rightly concluded that their study shows that, from a machine vision point of view, the difference between adulterated and unadulterated samples is mainly due to the difference in color or texture. They also rightly concluded that combining vision technology and chemometric methods is an effective method of identifying adulterated mutton, not only for various farm animals and poultry meat but also for adulterated meat.

Conclusion

It presents a complete research methodology, correctly selected methods, and quite broadly and objectively presents assumptions and weaknesses of the technique about the whole process of identifying adulterated mutton.

In subsequent studies, I suggest the inclusion of other modalities used in practice, such as, for example, spectrophotometry, which may also have an exciting application in the presented problem.

Author Response

(The authors gave the same response as above.)
